METHODS

# The NOSTRA model: Coherent estimation of infection sources in the case of possible nosocomial transmission

**David J. Pascall** [1]*, **Christopher Jackson**[1], **Stephanie Evans**[2], **Theodore Gouliouris**[3,4], **Christopher J.R. Illingworth**[5], **Stefan G. Piatek** [6], **Julie V. Robotham**[2,7], **Oliver Stirrup** [8], **Ben Warne**[3,9], **Judith Breuer**[10], **Daniela De Angelis**[1,11]

**1** MRC Biostatistics Unit, University of Cambridge, Cambridge, United Kingdom, **2** HCAI, Fungal, AMR, AMU & Sepsis Division, UK Health Security Agency, London, United Kingdom, **3** Clinical Microbiology and Public Health Laboratory, Cambridge University Hospitals NHS Foundation Trust, Cambridge, United Kingdom, **4** Clinical Microbiology and Public Health Laboratory, Cambridge, United Kingdom, **5** MRC-University of Glasgow Centre for Virus Research, University of Glasgow, Glasgow, United Kingdom, **6** Advanced Research Computing, University College London, London, United Kingdom, **7** NIHR Health Protection Research Unit in Modelling and Health Economics, Imperial College London, London, United Kingdom, **8** Institute for Global Health, University College London, London, United Kingdom, **9** Department of Medicine, University of Cambridge, Cambridge, United Kingdom, **10** Infection, Immunity and Inflammation Department, Great Ormond Street Institute of Child Health, University College London, London, United Kingdom, **11** Statistics, Modelling and Economics Department, UK Health Security Agency, London, United Kingdom

* david.pascall@mrc-bsu.cam.ac.uk

**Data availability statement:** The code for the analyses and the implementation of the

## Abstract

Nosocomial, or hospital-acquired, infections are a key determinant of patient health in healthcare facilities, leading to longer stays and increased mortality. In addition to the direct effects on infected patients, the burden imposed by nosocomial infections impacts both staff and other patients by increasing the load on the healthcare system. The appropriate infection control response may differ depending on whether the infection was acquired in the hospital or the community. For example, nosocomial outbreaks may require ward closures to reduce the risk of onward transmission, whilst this may not be an appropriate response to repeated importations of infections from outside the facility. Unfortunately, it is often unclear whether an infection detected in a healthcare facility is nosocomial, as the time of infection is unobserved. Given this, there is a strong case for the development of models that can integrate multiple datasets available in hospitals to assess whether an infection detected in a hospital is nosocomial. When assessing nosocomiality, it is beneficial to take into account both whether the timing of infection is consistent with hospital acquisition and whether there are any likely candidates within the hospital who could have been the source of the infection. In this work, we developed a Bayesian model which jointly estimates whether a given infection detected in hospital is nosocomial and whether it came from a set of individuals identified as candidates by hospital staff. The model coherently integrates pathogen genetic information, the timings of epidemiological events, such as symptom onset, and location data on the infected patient and candidate infectors. We illustrated this model on a real hospital dataset showing both

NOSTRA model is available at https://github.com/dpascall/NOSTRA-model. The Cambridge University Hospitals data used to illustrate the model are patient identifiable, and as such are not available. The data required to repeat the simulation study is available at https://doi.org/10.6084/m9.figshare.27172758.

**Funding:** DJP was funded by a NIHR award to JB (NIHR200652). CI was funded by the Medical Research Council (MC\UU\00034/1). JVR was supported by the National Institute for Health and Care Research (NIHR) Health Protection Research Unit (HPRU) in Modelling and Health Economics, which is a partnership between the UK Health Security Agency (UKHSA), Imperial College London, and the London School of Hygiene and Tropical Medicine (NIHR200908). This work was supported by UKRI through the JUNIPER consortium (MR/V038613/1). This work was supported by the Medical Research Council via the MRC Biostatistics Unit Core Award (MC\UU\00002/11). This work was supported by the NIHR Cambridge Biomedical Research Centre (NIHR203312). The views expressed are those of the authors and not necessarily those of the NIHR or the Department of Health and Social Care. The funders had no role in study design, data collection and analysis, decision to publish, or preparation of the manuscript.

**Competing interests:** The authors have declared that no competing interests exist.

its output and how the impact of the different data sources on the assessed probabilities are contingent on what other data has been included in the model, and validated the calibration of the predictions against simulated data.

## Author summary

Nosocomial, or hospital-acquired, infections have important consequences for patients and hospital staff: they worsen patient outcomes and their management stresses already overburdened health systems. Accurate judgments of whether an infection is nosocomial helps staff make appropriate choices to protect other patients within the hospital, as such appropriate models to assess whether an infection is nosocomial are a key public health need. Our assessed probability of nosocomiality should change if the infected patient came into contact with high-risk potential infectors within the hospital, and as such, we should not attempt to judge whether an infection is nosocomial without also considering this factor. Given this, we developed a model that integrates epidemiological, contact and pathogen genetic data to determine how likely an infection is to be nosocomial and the probability of given infection candidates being the source of the infection, and validated this model using simulations from a previously published agent-based hospital outbreak simulation model.

## Introduction

Nosocomial infections are an important issue facing health systems across the world, impacting both patient survival [1,2] and their willingness to access healthcare [3]. However, when an infection is found in a patient in a healthcare setting it is often unclear whether the infection was genuinely acquired in the facility or just detected there. These two scenarios have very different implications for decision making. High levels of nosocomial infections may require the stepping up of infection control responses both at the ward level, and across the whole hospital to protect patient health [4]. The precise required response may be idiosyncratic, as the ultimate causes of nosocomial infections differ dramatically across pathogens and facilities. For example, poorly ventilated wards may be susceptible to frequent nosocomial respiratory infections, while wards containing particularly susceptible patient groups may be at risk of high rates of nosocomial infection across pathogen groups. The ideal responses to these two cases will be very different, and the set of available responses will be limited by the economic context. But while the responses across scenarios and facilities will differ, in order to know whether they are necessary at all, it is first necessary to determine whether the infections that are being seen truly are healthcare-associated. Given this, methods for assessing both whether an infection is nosocomial, and if it is, whether it is part of a large hospital outbreak, are necessary for coherent decision making in infection control.

Historically, whether an infection was hospital acquired was commonly assessed by the application of heuristic criteria. For example, England's public health agency, the UK Health Security Agency (then Public Health England), in its last major study of nosocomial infections, used their main definition of a hospital-acquired infection as being an infection where symptom onset occurs on at least the 3rd day post-admission [5]. Moving beyond this heuristic approach to one based on specific evidence related to characteristics of the pathogen of

interest and host population would be more principled, with integrating larger amounts of data on a coherent fashion hopefully leading to more accurate assessments.

Tools have been developed explicitly for the assessment of nosocomiality [6–8] or the related task of inferring transmission histories (reviewed in [9]). However, these tools cannot simultaneously answer the questions of "Is the infection nosocomial?" and "What is the most likely source?". Treating both questions simultaneously should lead to systematically different answers than would be the case if attempting to answer them individually, because the presence or absence of high probability infection candidates within the hospital should impact our assessment of the probability that the infection is nosocomial. Given that the appropriate infection control response depends on whether an infection truly is nosocomial, there is an urgent need for the development of tools that are capable of simultaneously answering both of these questions, which can be used routinely by hospital staff in real time to guide management decisions.

Unfortunately, attributing nosocomiality and assessing infection sources of a given infection is a non-trivial task, as generally not all infection candidates will be identified, and even in cases where every possible infection candidate is known, data may only be available for a subset. Hence, any method to approach these problems together (or, indeed, individually) must decide what assumptions are going to be made about this problem of missing infection sources (and their associated data). This is done by either making assumptions about the nature of the missing data (commonly that data that is available is representative of the missing data), as in [6], or by reducing focus to a circumscribed question that is answerable with just the data observed, as in [10–12].

One model that directly attempted to assess nosocomiality was Hospital-Onset COVID-19 Infections (HOCI) [6]. This Bayesian model was developed at the height of the SARS-CoV-2 pandemic to integrate epidemiological and genetic data to assess whether detected SARS-CoV-2 infections were hospital-acquired. To model the epidemiological information, it used a distribution from infection time to symptom onset to assess whether infection was more likely in the hospital or the community given the observed admission and onset times. The genetics were modelled by testing the consistency of the viral isolate from the patient of interest to sequences from the hospital relative to those from the community. This tool was rolled out across National Health Service (NHS) hospitals in the UK for real time use by infection control teams in a large study assessing the impact of sequencing on clinical outcomes [13]. HOCI, however, did not attempt to make any assessments of the source of the infection if it was determined to have a high posterior probability of being nosocomial.

In contrast, the A2B model [11], focuses exclusively on source attribution. That is, conditional on there having been an infection in hospital, which individuals are consistent with having been the source. A2B works within a frequentist framework providing $p$-values on whether the data are more extreme than would be observed under the null model of infection from individual A to individual B, hence the name. Like HOCI, A2B makes use of infection times and sequence information, but it also uses information about the locations of individuals within the hospital, in order to rule out some transmission events.

Here, we, a group of authors comprising many of the original developers of the HOCI and A2B models, present a novel Bayesian model, designed as a conceptual integration of these two models [6,11], that we call NOSTRA (NOSocomial TRansmission Assessment), a name inspired by the historical prognosticator Nostradamus. The aims of our model are twofold:

- To provide both a probability a detected infection was acquired within the hospital, and probabilities for the source being within a set of given candidate individuals

- To have a low enough runtime to be usable on wards in real-time for clinical decision making

We illustrate the outputs of this model using previously published data [10] collected during the early stages of the COVID-19 pandemic at Cambridge University Hospitals NHS Foundation Trust (CUH) and validate its performance against infection data from an agent-based hospital simulation model.

## Materials and methods

### Data

The data we used to illustrate our model outputs are fully explained in Illingworth et al. [10], but we briefly re-describe it here. The data were collected during prospective COVID-19 surveillance at CUH between the 22nd March and 14th June 2020. Patients were tested for COVID-19 through targeted patient screening in wards with detected hospital-onset outbreaks. The original data comprised five wards, but we focus only on the ward identified as A in Illingworth et al. Due to a large outbreak on this ward, all individuals on the ward were eventually tested, including those exhibiting no symptoms. Final case sets were generated manually by seeking for possible links in a social network diagram generated in FoodChain-lab [14].

Patient locations on each day and the date of onset of symptoms (or in the case of asymptomatic infections, date of detection) were extracted from the hospital's electronic records. Location data were available for all but two patients.

Viral sequences were generated from isolates using the modified ARTIC v2 protocol [15]. These sequences represent the string of nucleic acids that make up the viral genome, and differences between them can be informative on how the viral populations within different patients are related to one another. The whole viral population in a patient is summarised as a "consensus sequence," which can be viewed as the average viral genome in that patient.

For this study, we performed some post-processing of the genetic data, firstly reducing the alignment for each pair of individuals to those columns which contain no ambiguities or gaps, recorded the length of this reduced alignment, and calculated the number of single nucleotide polymorphisms (SNPs) between each pair in the reduced alignment.

### Model

The goal of the model is to estimate the most likely source of infection for an individual, who we label $B$, whose infection was discovered in hospital using: data on $B$'s movements; the genetic sequence of their infecting pathogen; their times of admission to hospital and symptom onset; and the movements and onset times of a set of $n$ candidate infector individuals in the hospital, who we label $A_1$ to $A_n$. We partition the possible sources of infection into $n + 2$ mutually exclusive groups as follows:

- the candidate individuals, labelled $A_1, \ldots, A_n$,
- any source of infection from the hospital other than these individuals, including visitors, labelled $H$,
- infections outside the hospital, labelled $C$.

We set this up a Bayesian inference problem. The unknown source of infection $S$ is a categorical variable with $n + 2$ potential values, $s \in \{A_1, \ldots, A_n, H, C\}$. The goal is to estimate the

posterior distribution of $S$. This posterior distribution is fully specified by the set of quantities $P(S = s|X) : s \in \{A_1, ...A_n, H, C\}$, with $\sum_s P(S = s|X) = 1$, and expresses our judgment and uncertainty about $B$'s true infection source after observing the data, $X$. For example, if we became certain that $B$ was infected outside the hospital, we would have $P(S = C|X) = 1$, and a probability of 0 for each other potential source. We start with a prior probability distribution $P(S = s) : s \in \{A_1, ...A_n, H, C\}$ and deduce the posterior distribution given this prior and the observed data. All notation is defined in Table 1.

We partition the data $X$ into several components, such that $X = \{X_{A_1}, ..., X_{A_n}, X_H\}$. $X_{A_z}$ consists of the onset time of $A_z$, $t_o^{A_z}$, the distance in terms of SNPs between their pathogen

**Table 1. A reference for the notation used throughout the paper.**

| Notation | Definition |
|---|---|
| $B$ | The focal individual of the study whose source of infection is to be determined |
| $S$ | The set of $n + 2$ infection sources |
| $A_z$ | The infection source consisting of the $z$th candidate individual within the hospital for the infection of $B$ |
| $H$ | The infection source for the infection of $B$ consisting of all unknown individuals in the hospital, including visitors |
| $C$ | The infection source for the infection of $B$ consisting of all individuals outside of the hospital |
| $X$ | All available data |
| $X_{A_z}$ | The portion of the data containing the symptom onset time of $A_z$, the distance in terms of single nucleotide polymorphisms (SNPs) between their pathogen genomes, and the set of days that $A_z$ and $B$ were at the in the same location |
| $X_B$ | The portion of the data containing the hospital admission time of $B$, and the symptom onset time of $B$ |
| $L(X|Z)$ | The likelihood of data $X$ given parameters $Z$ |
| $P(H = h|Z)$ | The conditional probability or conditional probability density of the realisation $h$ of the random variable $H$ given $Z$ |
| Distribution$(X|Z)$ | The probability or probability density of data $X$ under the given distribution with parameters $Z$ |
| $T_o^I, t_o^I$ | The random variable describing the symptom onset time of individual $I$, and its realisation, respectively |
| $T_i^I, t_i^I$ | The random variable describing the infection time of individual $I$, and its realisation, respectively |
| $T_w^I, t_w^I$ | The random variable describing time between infection and symptom onset of individual $I$, and its realisation, respectively |
| $T_{MRCA}^{I,J}$ | The random variable describing time in viral generations since the viral isolates from individuals $I$ and $J$ shared a common ancestor |
| $t_s^I$ | The sampling time of $I$'s pathogen isolate |
| $t_d^{I,J}$ | The absolute time difference between the sampling time of $I$'s pathogen isolate and $J$'s pathogen isolate, that is $|t_s^I - t_s^J|$ |
| $t_a^I$ | The hospital admission time of individual $I$ |
| $t_\varnothing$ | The start time of the epidemic (or simulation) |
| $t_e^n$ | The time of the first event in the $n$th transmission tree |
| $\Delta^{I,J}, \delta^{I,J}$ | The random variable describing the number of SNPs between the viral isolates from individuals $I$ and $J$, and its realisation |
| $D^{I,J}, d^{I,J}$ | The random variable describing whether individuals $I$ and $J$ were in contact on a set of days, and its realisation |
| $E$ | The per base error probability of the sequencing technology employed to generate the pathogen genome sequences |
| $G$ | The genome length of the pathogen under study |
| $G^{I,J}$ | The effective genome length of the alignment between individuals $I$ and $J$ after removing gaps and ambiguities in the alignment |
| $N_e$ | The effective population size of the pathogen under study at the time of sampling |
| $M$ | The evolutionary rate of the pathogen under study, in mutations per unit time |
| $g$ | The mean generation time of the pathogen under study |

genomes (assumed to be generated from an alignment with no gaps or ambiguities), $\delta^{A_z,B}$, and whether $A_z$ and $B$ were in the in the same location on each day, $d^{A_z,B}$. $X_H$ consists of the admission time of $B$, $t_a^B$, and the onset time of $B$, $t_o^B$.

## Bayesian analysis: prior

Bayesian analysis requires a prior over $S$. There are multiple ways that this prior could be justified. Our default prior is a uniform prior over nosocomiality. More precisely, this prior is of the form $P(S = C) = 0.5$, $P(S = s) = \frac{1}{2n+2}$ for $s \in \{A_1, ..., A_n, H\}$, where $n$ is the number of candidate individuals. This prior expresses complete uncertainty over whether a given infection is nosocomial, and, conditional on it being nosocomial, complete uncertainty over its source within the healthcare facility.

## Bayesian analysis: likelihood

For Bayesian estimation of $P(S|X)$, we need to define the likelihood of the data $X$ for each potential value (or "hypothesis") for the unknown $S$. We denote this $L(X|S = s)$, and we now define it in turn for each $s$.

There are two broad classes of hypothesis; when the infection is not from a candidate individual ($S \in \{H, C\}$) and when it is ($S \in \{A_1, ..., A_n\}$). Within each class of hypothesis, the likelihood has the same general structure. We will treat them one at a time.

**Likelihood of the data given that infection was from a non-candidate in the hospital or in the community,** $L(X|S \in \{H, C\})$   Under this class of hypothesis, the infection occurred in the community or from an unknown individual in the hospital.

We make the strong simplifying assumption that each of the components of $X$ are generated independently of one another, and hence the likelihood factorises. This assumption rules out indirect transmission between candidate individuals and the focal individual, as in the scenario $B$'s infection came indirectly from $A_z$ via a person in $H$ or $C$, the onset time of $A_z$ would not be independent of that of $B$.

Under this assumption:

$$L(X|S = H) = L(X_H|S = H) \prod_{z=1}^{n} L(X_{A_z}|S = H) \tag{1}$$

and

$$L(X|S = C) = L(X_H|S = C) \prod_{z=1}^{n} L(X_{A_z}|S = C) \tag{2}$$

We will now derive each component of these likelihoods separately.

**Likelihood of $X_H$ given that infection was in the community,** $L(X_H|S = C)$   This is the likelihood for the onset time of person B, $t_o^B$, given their admission time, $t_a^B$. This is obtained by specifying parametric models for B's (unknown) infection time $T_i^B$, assumed to have density $f()$, and the incubation time between B's infection and onset, $T_w^B = T_o^B - T_i^B$, assumed to have density $g()$.

Suppose we knew the infection time was $t_i^B$, then the onset time is $t_o^B = t_i^B + t_w^B$. The probability of observing $t_o^B$ could then be obtained directly from the model for $T_w^B$, that is, $P(T_w^B = t_o^B - t_i^B)$. However, we do not know the infection time, so the likelihood for $t_o^B$ is determined by integrating this probability over the range of values of $t_i^B$ compatible with having acquired the infection outside of hospital, that is, infection times between the start of the epidemic $t_\varnothing$ and

the admission time $t_a^B$:

$$L(X|S=C) = P(T_o^B = t_o^B|S=C) = \int_{t_\varnothing}^{t_a^B} P(T_w^B = t_o^B - t_i^B)P(T_i^B = t_i^B)dt_i^B \tag{3}$$

**Likelihood of $X_H$ given that infection was from a non-candidate in the hospital, $L(X_H|S=H)$** This is as $L(X_H|S=C)$, except that the integral is taken over the range of infection times that are compatible with infection being acquired from an unidentified individual within the hospital between $t_a^B$ and $t_o^B$.

**Likelihood of $X_{A_z}$ given that infection was from a non-candidate in the hospital or in the community, $L(X_{A_z}|S \in \{C,H\})$** $X_{A_z}$ contains the onset time of $A_z$, $t_o^{A_z}$, the difference in terms of SNPs between the sequenced genomes of the pathogens infecting $A_z$ and $B$, $\delta^{A_z,B}$, a vector of 1s and 0s describing on which $A_z$ and $B$ were in the same location, $d^{A_z,B}$, and for any unobserved elements of $d^{A_z,B}$ elicited probabilities that $A_z$ and $B$ were in contact on those days, $w(A_z,B)$. Note that, as none of these data are impacted by whether $B$ was infected in the hospital or the community, the likelihood is identical under both $S = H$ and $S = C$. Under these hypotheses, $A_z$ was not the source of $B$'s infection, hence, we assume that the components of $A_z$'s data ($A_z$'s onset time, the genetic difference between the viruses affecting $A_z$ and $B$, and $A_z$'s co-location with $B$) are independent of each other. As before, this independence is plausible if there was no intermediate transmission between $A_z$ and $B$. This implies that $A_z$ did not infect $B$ (and vice versa), their co-locations, $d^{A_z,B}$, are independent of the other data in $X_{A_z}$ and the genetic data is independent of the epidemiological data and thus can be modelled separately.

Given this, we have:

$$\begin{aligned} L(X_{A_z}|S \in \{C,H\}) &= P(\Delta^{A_z,B} = \delta^{A_z,B}, T_o^{A_z} = t_o^{A_z}, D^{A_z,B} = d^{A_z,B}|S \in \{C,H\}) \\ &= P(\Delta^{A_z,B} = \delta^{A_z,B}|S \in \{C,H\})P(T_o^{A_z} = t_o^{A_z}|S \in \{C,H\}) \\ &\quad P(D^{A_z,B} = d^{A_z,B}|S \in \{C,H\}) \end{aligned} \tag{4}$$

where $\Delta^{A_z,B}$ and $D^{A_z,B}$ are the random variables underlying the observed genetic data $\delta^{A_z,B}$ and co-location data $d^{A_z,B}$.

We already have a parametric model for onset time, which we applied to $B$'s onset time above. The same model can be applied to $A_z$'s onset time, with the integration for $t_i^{A_z}$ being over the possible infection dates for $A_z$, that is between $t_\varnothing$ and $t_o^{A_z}$.

For the co-locations, $d^{A_z,B}$, we follow the approach taken in A2B [11]. Individuals either are or are not in contact on any particular day, giving a total of $2^{|D|}$ potential contact history vectors for $|D|$ days. Assuming that none of these contact histories are more or less likely than any other given $A_z$ did not transmit to $B$, the observed contact history then has probability $P(D^{A_z,B} = d^{A_z,B}|S \in \{C,H\}) = 0.5^{|D|}$.

To specify $P(\Delta^{A_z,B} = \delta^{A_z,B}|S \in \{C,H\})$ we make use of the coalescent [16–18]. All the necessary genealogical theory for this section is reviewed in Hudson 1990 [19]. Assume that the viruses are evolving under a Poisson process with rate, $M$. Under the coalescent, the number of generations to the most recent common ancestor (MRCA), for two randomly chosen individuals, is exponentially distributed with rate given by the inverse of the (effective) population size, $N_e$. Let $T_{MRCA}^{A_z,B}$ represent this random variable. Since this represents the number of generations since the pathogens infecting $A_z$ and $B$ last shared a common ancestor, the pathogens are separated by $2T_{MRCA}^{A_z,B}$ generations of independent evolution at rate $M$. Hence, during the time in standard units spanned by these generations, we would

expect the number of SNPs generated through evolution to be Poisson distributed with mean $2T_{MRCA}^{A_z,B}MgG$.

Our assumption that the alignment has no gaps or ambiguities is unrealistic, so we create an accounting variable for each candidate individual and the focal individual $G^{A_z,B}$, which corresponds to the effective genome size after ambiguous sites and gaps have been removed. We assume that the alignment of $G^{A_z,B}$ length is comparable to the unrealised complete alignment of length $G$. We use this new variable to correct the mean to $2T_{MRCA}^{A_z,B}MgG^{A_z,B}$.

As $T_{MRCA}^{A_z,B}$ is an exponentially distributed random variable with rate $1/N_e$, $2T_{MRCA}^{A_z,B}MgG^{A_z,B}$ is also exponentially distributed with rate, $\frac{1}{2MgN_eG^{A_z,B}}$. As a Poisson distribution with a Gamma-distributed random rate parameter is equivalent to a Negative Binomial distribution, the number of mutations generated through evolution between the two sequences can be modelled as $NB[r = 1, p = 1/(1 + 2MgN_eG^{A_z,B})]$.

In addition, there would then be differences added by sequencing error in both genomes. We can model the number of sequencing errors as a Binomial random variable with probability $E$, the per base error probability and number of trials $2G$, double the genome size, as this occurs in both genomes. As $E$ will be small and $2G$ is large, we approximate this Binomial distribution with a Poisson distribution with rate $2EG$. Again, we correct for the observed genome length by modifying this rate to $2EG^{A_z,B}$.

Therefore, the total number of genetic differences between the two genomes is the sum of the Negative Binomial distribution describing the SNPs generated though mutation and the Poisson approximation to the Binomial distribution describing the SNPs generated through sequencing error (assuming no back mutation). The sum of a Negative Binomial distributed random variable and a Poisson distributed random variable is Delaporte distributed [20]. Hence, the likelihood is:

$$
\begin{aligned}
&P(\Delta^{A_z,B} = \delta^{A_z,B}|S \neq A_z) \\
&\quad = \text{Delaporte}\left(\delta^{A_z,B}|\alpha = 2MgN_eG^{A_z,B}, \beta = 1, \lambda = 2EG^{A_z,B}\right)
\end{aligned}
\tag{5}
$$

Note that if the isolates are collected on different days with time difference in standard units, $t_d^{A_z,B}$, the distribution of the time between them would be $\text{Exp}(\frac{1}{2gN_e}) + t_d^{A_z,B}$ instead of just $\text{Exp}(\frac{1}{2gN_e})$. This can be accounted for by modifying the $\lambda$ parameter of the Delaporte distribution from $\lambda = 2EG^{A_z,B}$ to $\lambda = G^{A_z,B}\left(2E + \frac{t_d^{A_z,B}M}{G}\right)$.

**Likelihood of the data given that infection was from a candidate individual, $L(X|S \in \{A_1,...,A_n\})$**   Under this class of hypothesis, $B$ was infected by one of the candidate individuals $A_z$. We assume then that the data $X_{A_j}$, that describe the relationship between $B$ and each other individual $A_j, j \in \{1,...,n \quad \backslash \quad z\}$, are independent between each $A_j$, and independent of the data $X_{A_z}$ that describe the relationship between the infecting $A_z$ and $B$. That is:

$$
L(X|S = A_z) = L(X_H, X_{A_z}|S = A_z)\prod_{j=1,j\neq z}^{n} L(X_{A_j}|S = A_z)
\tag{6}
$$

$L(X_{A_j}|S = A_z)$ takes the same form as $L(X_{A_z}|S \in \{C, H\})$. All that remains is to generate a parametric model for $L(X_H, X_{A_z}|S = A_z)$.

**Likelihood of $X_H$ and $X_{A_z}$ given that infection was from the candidate individual $A_z$, $L(X_H, X_{A_z}|S = A_z)$**   We partition the data into event times $T$, genetic distance $\Delta$, and co-location $D$ components, and rearrange in terms of the conditional probability of $B$'s data given

$A_z$'s onset time $T_o^{A_z}$.

$$L(X_H, X_{A_z}|S = A_z) = P(\Delta^{A_z,B} = \delta^{A_z,B}, T_o^{A_z} = t_o^{A_z}, T_o^B = t_o^B, D^{A_z,B} = d^{A_z,B}|S = A_z)$$
$$= P(\Delta^{A_z,B} = \delta^{A_z,B}, T_o^B = t_o^B, D^{A_z,B} = d^{A_z,B}|T_o^{A_z} = t_o^{A_z}, S = A_z) \qquad (7)$$
$$P(T_o^{A_z} = t_o^{A_z}|S = A_z)$$

$P(T_o^{A_z} = t_o^{A_z}|S = A_z)$ has been defined above.

We then obtain $P(\Delta^{A_z,B} = \delta^{A_z,B}, T_o^B = t_o^B, D^{A_z,B} = d^{A_z,B}|T_o^{A_z} = t_o^{A_z}, S = A_z)$ using the same technique as in the A2B model [11]. This involves expanding this term by summing it over $B$'s unknown time of infection $T_i^B$, and assuming that $B$'s onset time, the genetic distance of $B$'s pathogen from $A_z$'s, and $B$'s co-location with $A_z$ are conditionally independent given this infection time. Specifically:

$$P(\Delta^{A_z,B} = \delta^{A_z,B}, T_o^B = t_o^B, D^{A_z,B} = d^{A_z,B}|T_o^{A_z} = t_o^{A_z}, S = A_z) =$$

$$\sum_{t=t_\varnothing}^{t_o^B} P(T_i^B = t|T_o^{A_z} = t_o^{A_z}, S = A_z)P(T_o^B = t_o^B|T_i^B = t, T_o^{A_z} = t_o^{A_z}, S = A_z) \qquad (8)$$

$$P(\Delta^{A_z,B} = \delta^{A_z,B}|T_i^B = t, T_o^{A_z} = t_o^{A_z}, S = A_z)$$
$$P(D^{A_z,B} = d^{A_z,B}|T_i^B = t, T_o^{A_z} = t_o^{A_z}, S = A_z)$$

Furthermore, we note that the onset time of $A_z$ provides no extra information after conditioning on the infection time of $B$, so we can simplify as follows:

$$P(T_o^B = t_o^B|T_i^B = t, T_o^{A_z} = t_o^{A_z}, S = A_z) = P(T_w^B = t_o^B - t) \qquad (9)$$

$$P(\Delta^{A_z,B} = \delta^{A_z,B}|T_i^B = t, T_o^{A_z} = t_o^{A_z}, S = A_z) = P(\Delta^{A_z,B} = \delta^{A_z,B}|T_i^B = t, S = A_z) \qquad (10)$$

$$P(D^{A_z,B} = d^{A_z,B}|T_i^B = t, T_o^{A_z} = t_o^{A_z}, S = A_z) = P(D^{A_z,B} = d^{A_z,B}|T_i^B = t, S = A_z) \qquad (11)$$

Any distributional form could be assumed for $P(T_i^B = t|T_o^{A_z} = t_o^{A_z}, S = A_z)$ and this choice should be specific to the pathogen in question and informed from its epidemiological literature.

For the term $P(\Delta^{A_z,B} = \delta^{A_z,B}|T_i^B = t, S = A_z)$, we take a similar the approach for the genetics used for $L(X_{A_z}|S \in \{C, H\})$, but the situation is drastically simplified as the infection time is known. We make the assumption of no within host variation in the pathogen, so that there is no risk of incomplete lineage sorting and the time of the MRCA of $B$ and $A_z$ is exactly the infection time of $B$. If $t$ is the infection time of $B$, and $t_s^B$ and $t_s^{A_z}$ are the sampling times of the pathogen genomes of $B$ and $A_z$ respectively, then there has been $|t_s^B - t| + |t_s^{A_z} - t|$ time units of independent evolution for the pathogens. Given the Poisson process assumption for mutation being used, we expect the number of SNPs between the genomes to follow a Poisson distribution with mean $(|t_s^B - t| + |t_s^{A_z} - t|)MG$, which after accounting for partial observation, becomes $(|t_s^B - t| + |t_s^{A_z} - t|)MG^{A_z,B}$. Again, we assume that additional mutations are generated by sequencing error, following a Poisson distribution with mean $2EG^{A_z,B}$. As the sum of two Poisson random variables is Poisson, this gives that:

$$P(\Delta^{A_z,B} = \delta^{A_z,B}|T_i^B = t, S = A_z) =$$
$$\text{Poisson}\left(\delta^{A_z,B}|\lambda = (|t_s^B - t| + |t_s^{A_z} - t|)MG^{A_z,B} + 2EG^{A_z,B}\right) \quad (12)$$

Finally, consider the likelihood term $P(D^{A_z,B} = d^{A_z,B}|T_i^B = t, S = A_z)$ for the history of co-location $d^{A_z,B}$ of individuals $A_z$ and $B$ under a hypothesis that $A_z$ infected $B$ at time $t$. These data consists of a vector of $d^{A_z,B}$, where $d_c^{A_z,B} = 1$ or $d_c^{A_z,B} = 0$ if $A_z$ and $B$ are known to have been co-located, or not co-located, respectively, for each day $c$. On some occasions $d_c^{A_z,B}$ is unknown, and we assume we have an elicited probability $w_c(A_z, B)$ that they were in contact then. In the case of fully observed location data, $w(A_z, B)$ is undefined.

Then, assuming conditional independence of the co-location data on each day, we have:

$$P(D^{A_z,B} = d^{A_z,B}|T_i^B = t, S = A_z) = \prod_c p_c(A_z, B, t) \tag{13}$$

where

$$p_c(A_z, B, t) = P(D_c^{A_z,B} = d_c^{A_z,B}|T_i^B = t, S = A_z) \tag{14}$$

at times $c$ when $d_c^{A_z,B}$ is observed. When contact status is unobserved, this likelihood contribution is defined as a weighted average over the missing contact indicator $d_c^{A_z,B}$, using the estimated contact probabilities as weights, giving:

$$p_c(A_z, B, t) = (1 - w_c(A_z, B))P(D_c^{A_z,B} = 0|T_i^B = t, S = A_z) + \\ w_c(A_z, B)P(D_c^{A_z,B} = 1|T_i^B = t, S = A_z) \tag{15}$$

To specify these likelihood contributions, firstly note that if $T_i^B = t$, then $A_z$ and $B$ must have been in contact at day $t$, since contact is necessary for infection. Therefore if $c = t$, then $P(D_c^{A_z,B} = d_c^{A_z,B}|T_i^B = t, S = A_z) = p_c(A_z, B, t)$. At any other time $c$ (following Illingworth et al. [11]) we suppose that all observed co-location patterns are equally plausible, implying that $P(D_c^{A_z,B} = d_c^{A_z,B}|T_i^B = t, S = A_z) = 0.5$.

## Bayesian analysis: inference

With the likelihood and priors defined as above, the posterior probabilities of the different categorical sources of infection are available analytically, and can simply be calculated directly, without recourse to any kind of numerical integration, such as MCMC. That is:

$$P(S = s|X) = \frac{L(X|S = s)P(S = s)}{\sum_{j \in \{A_1,..,A_n,H,C\}} L(X|S = j)P(S = j)} \tag{16}$$

This gives us the posterior probability of each of the possible infection sources being the source of the infection. The community probability, $P(S = C|X)$, is the probability the infection was obtained outside the medical facility. The nosocomiality probability is $1 - P(S = C|X)$, that is, the sum of all the non-community probabilities. The probability of the hospital compartment, $P(S = H|X)$, is the probability that the infection came from a source within the medical facility that had not been *a priori* identified by the medical staff.

Note that, under NOSTRA, if no candidate infectors are provided the posterior is purely a function of the length of the waiting time between admission and symptom onset/detection:

$$P(S = H|X) = \frac{L(X_H|S = H)P(S = H)}{L(X_H|S = H)P(S = H) + L(X_H|S = C)P(S = C)} \tag{17}$$

## Model illustration

This section briefly states which distributions and parameters we used for our illustration of the model with the CUH SARS-CoV-2 data. We use the default prior described in the prior section as the prior on $S$. $T_i^B$ is a given a uniform distribution between $t_o^B$ and $t_\varnothing$. $T_i^{A_z}$ is a given a uniform distribution between $t_o^{A_z}$ and $t_\varnothing$. Following Illingworth et al. [11,21], we give $T_w^B$ a lognormal distribution with mean = 1.434 and standard deviation = 0.6612, and approximate $2EG^{A_z,B}$ with 0.404. We follow Ferretti et al. [22] and give $P(T_i^B = t | T_o^{A_z} = t_o^{A_z}, S = A_z)$ a shifted Student's t-distribution with shift = -0.078, scale = 1.857, and df = 3.345, and set the generation time, $g$, to 5.5. We took the mutation rate $M = 1.829 \times 10^{-6}$ from Wang et al. 2022 [23] and set $N_e = 51$ based off an approximation from their figure. We set $t_\varnothing$ to the 30th December 2019, the earliest admission time for a patient on the ward.

## Simulation validation

We performed a simulation analysis to validate this method. We used a SARS-CoV-2 calibrated individual-based hospital simulation model previously generated by some of the authors [24] to generate full infection histories of infections detected in hospital, some of which have are acquired in the community and some of which are acquired from patients or healthcare workers in the hospital. As we have the full infection histories of these patients, we know what the true infection source is. All data required for NOSTRA other than the genetics is generated during the process of running the model.

The transmission trees implicitly generated by the model are only for infections that occurred within the hospital itself, so the result is a set of disconnected graphs, with each graph corresponding to the hospital outbreak from a single imported SARS-CoV-2 infection. In order to simulate required genetics, we assumed that mutation occurred independently across each transmission tree following a Poisson process with rate $6.677 \times 10^{-4}$ yr$^{-1}$ site$^{-1}$. We drew the number of SNPs between each transmission tree from a Delaporte $(\alpha = 2MgN_eG, \beta = 1, \lambda = M(t_e^1 + t_e^2 - 2t_\varnothing))$ distribution, with $N_e = 51$, $g = 5.5$ days, $M = 6.677 \times 10^{-4}$ yr$^{-1}$ site$^{-1}$, $G = 29811$, $t_\varnothing$ being the starting time of the simulation, and $t_e^n$ being the time of the first event in the $n$th transmission tree.

We simulated the data under three broad conditions; high (6% community prevalence, 0.15% of new admissions infected), intermediate (4% community prevalence, 1% of new admissions infected) and low (2% community prevalence, 0.05% new admission infected) infection prevalence. Within each condition, simulations were executed for each of 20 calibrated hospital infection parameter sets (as described in Evans et al. 2021 [25], see S1 Table for precise parameter sets used), giving a total of 60 simulations. The hospital in each simulation has 42 wards.

NOSTRA was then run with and without candidate individuals on the data at the ward level on patients with detected infections admitted after the 50th day of the simulation (to allow the model to equilibrate) using the same parameter values reported above for the illustration. We used our advised default prior of $P(C) = 0.5$, with the rest of the probability divided equally between the rest of the sources. This represents complete uncertainty as to whether an infection is nosocomial or not. We assessed the model posterior against three references, the above prior, the (usually unknown) true prevalence of nosocomiality, and a rule of thumb where every detection 96 hours post-admission is assigned a nosocomial probability of 1 and every detection prior to this is assigned a nosocomial probability of 0. The true prevalence condition corresponds to a prior of the form $P(C) = \alpha$, where $\alpha$ is equal to the true proportion of cases after the 50th day of the given simulation that were actually acquired in

the community. The remaining $1 - \alpha$ of the probability is divided equally between the rest of the sources.

To quantify the quality of the estimates we used Brier scores [26], which measure the calibration of probabilistic forecasts. This was done on three targets; the nosocomiality itself, and the true infection source, and individuals within the same transmission chain as the true infection source. This allows the differential assessment of skill at whether an infection is nosocomial but not at where in the hospital it came from, or vice versa. The transmission chain test was performed because we expect, from the structure of the model, that it may inappropriately assign high probability of sourcehood to individuals who are in the same transmission chain of the true infector when that infector is not detected, given that the data for those individuals will look very like that of the true infector. As such, the transmission tree simulation will test whether the correct chains of transmission are being detected by the model. The tests were done on the full dataset of patients admitted after the 50th day, and for each subset of patients whose time difference between admission and detection were 0 to 9 and greater than 9 days. These subsets should represent different degrees of challenge for the model, with, for example, those being detected on the day they are admitted being easily identifiable as community-acquired, and the hardest cases being those detected a day or two after admission, where both within hospital and external infection are plausible. We used one-sided Bonferroni-corrected Wilcoxon signed rank tests to compare the performance of NOSTRA to the references used for the Brier scores from the full dataset. The use of the one-sided test is justified as we are only interested in whether NOSTRA performs better than the references. The rule of thumb and NOSTRA without candidate infectors references were only used in the nosocomial assessment comparison.

As analyses were run at the ward level, and all identified candidate individuals were therefore from the same ward, if the true infection source was in the hospital, but resulted from a cross-ward transmission chain, the truth was set as the $H$ source. Probabilities were converted from the per individual source level to the transmission tree level by summing over the individuals involved in each transmission tree.

## Results

### Model illustration

NOSTRA provides an estimate of the posterior probability for all of the possible infection sources (Fig 1), with rows corresponding to focal individuals and columns to infection sources. The probability of nosocomial infection, shown in the final column, is then the sum of the probabilities over the candidate individuals and hospital component.

As NOSTRA is capable of running with only symptom dates, we can assess the impact of each of the data sources on the posterior by adding one at a time. This is shown in Fig 2 where the change in posterior probability of the infection sources as different data is added is visible. We exclude patients CAMP000676 and CAMP000706, as their admission times and location data were unavailable. In this dataset, the genetics has a very large impact on the generated posterior probabilities. This appears to be driven by the genetic consistency of specific candidate individual's viral isolates to the focal individual's viral isolate dramatically reducing the posterior probabilities of the hospital and community compartments. For this dataset, the impact of the location information on the posterior is contingent on the data already in the model, with it only causing a large change in posterior probabilities if the genetics has already been added. This is because the strength of the location data is to rule out transmission by identifying potential transmission pairs who never interacted at the appropriate time and thus are very unlikely to have been linked. Hence, the location data has the largest impact when it

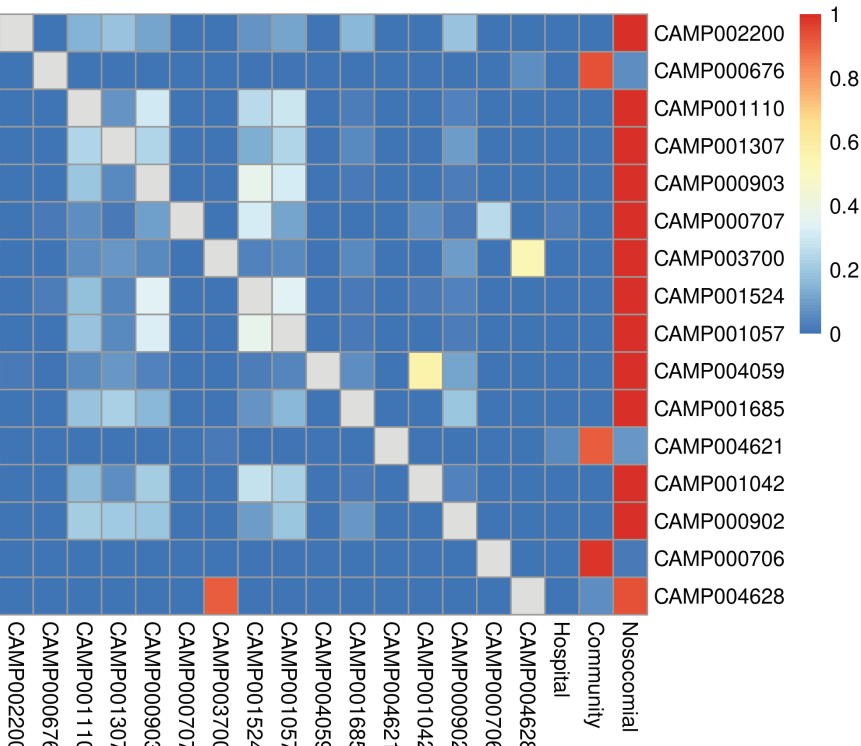

**Fig 1. A visualisation of the output of the model applied to the full CUH Ward A dataset.** Each row corresponds to a candidate individual, each column, except the last, to a potential infection source. Sources starting with "CAMP" are identified patients within the hospital. The "Hospital" source represents all unidentified sources within the hospital. The "Community" source is all sources outside the hospital. Cells are coloured by the posterior probability of that infection source. The last column shows the posterior nosocomiality probability, which is 1−$P$(Community|Data).

removes infection sources that were favoured by the genetics. Another example of the concentration of the posterior around specific sources as data more data is added is shown for analysis of one the simulation outputs in S1 Fig.

## Model validation

Fig 3 and Table 2 summarise NOSTRA's calibration for the different targets compared against a series of references; nosocomiality, source and transmission chain. The simulation prevalence has no consistent impact on the performance, though there is substantial variability among prevalence-parameter set combinations. NOSTRA performs well at nosocomial prediction with a mean absolute error in probability of 0.102, with it significantly exceeding the calibration of all references, including the unknown true frequency of nosocomiality. There is a clear gain in skill for nosocomial assessment from the joint estimation of the infection sources and nosocomiality relative to infectious sources alone. To illustrate this, the NOSTRA model which doesn't include candidates performs worst among all comparisons while the NOSTRA model including the candidates performs best. With respect to source attribution, NOSTRA performs significantly better that the comparator references, but only by a small amount. In the transmission chain task, NOSTRA performs comparatively to its ability at nosocomial prediction despite the extra difficulty. Given this, it appears that the poor source

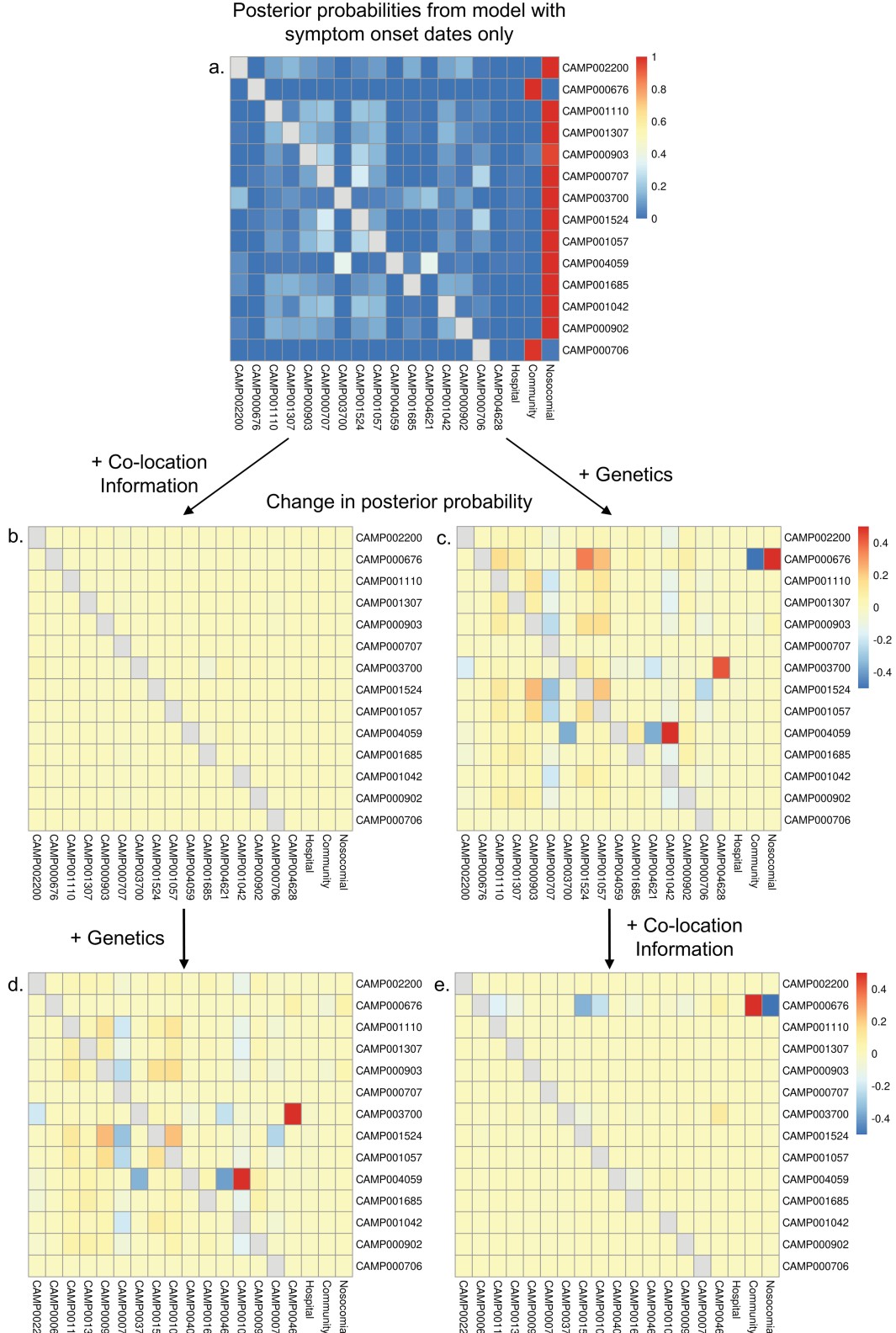

**Fig 2. The impact of adding data sources on the assessed probability of each infection source.** Panel a. shows the posterior probabilities of each infection source when only symptom onsets and admission times are provided to NOSTRA. The other panels explore the impact on the posterior probabilities as different data is added. Panel b. shows the change in posterior probability from the estimates in panel a. when location information is added. Panel c. shows the change in posterior probability from the estimates in panel a. when genetic information about the pathogen is added. Panel d. shows

the change in posterior probability from the model including onset times, admission times and patient locations, when genetic information about the pathogen is added. Panel e. shows the change in posterior probability from the model including onset times, admission times and genetic information about the pathogen, when patient locations are added.

**Table 2. The arithmetic mean, and 0.025 and 0.975 quantiles of the Brier scores of the different models for each of the different estimation targets NOSTRA, Candidates is the NOSTRA model run with a full set of candidate individuals and all data. NOSTRA, No Candidates is the NOSTRA model run with no candidate individuals using Eq 17. Prevalence Prior sets the prior probability of nosocomiality to the true probability of nosocomiality in the dataset. Naïve prior sets the prior probability of community infection to 0.5 and the probability of every source within the hospital to $\frac{1}{2n+2}$, where $n$ is the number of candidate individuals in the hospital. 96hr Categorisation assigns a nosocomiality probability of 0 to anything detected in the first 96 hours post-admission and a nosocomiality probability of 1 to everything else.**

| Nosocomiality Detection | |
|---|---|
| **Model** | **Mean Brier score (0.025 and 0.975 quantiles)** |
| NOSTRA - Candidates | 0.20 (0.12, 0.28) |
| NOSTRA - No Candidates | 0.82 (0.61, 1.02) |
| Prevalence Prior | 0.27 (0.07, 0.48) |
| Naïve Prior | 0.50 (0.50, 0.50) |
| 96hr Categorisation | 0.73 (0.54, 0.93) |
| **Source Detection** | |
| **Model** | **Mean Brier score (0.025 and 0.975 quantiles)** |
| NOSTRA - Candidates | 0.92 (0.77, 1.04) |
| Prevalence Prior | 0.94 (0.84, 0.99) |
| Naïve Prior | 1.06 (0.85, 1.21) |
| **Transmission Tree Detection** | |
| **Model** | **Mean Brier score (0.025 and 0.975 quantiles)** |
| NOSTRA - Candidates | 0.21 (0.13, 0.29) |
| Prevalence Prior | 0.91 (0.81, 0.97) |
| Naïve Prior | 1.03 (0.83, 1.19) |

attribution calibration is predominantly due to putting large amounts of posterior probability on candidate sources within the same transmission chain as the true source.

The waiting time from admission to detection is related to the difficulty in assigning its nosocomial status. When infection is detected on the day of admission, it is a simple task to determine that the infection was acquired in the community, and, likewise, if the infection is detected several weeks after admission, there is little doubt that an infection is nosocomial. Therefore, the most difficult cases are those that occur in the period consistent with both hospital and community acquisition. Fig 4 summarises the calibration of the model under different times from admission to detection. NOSTRA performs comparatively well to its average performance on the complete dataset (Fig 3) over this entire time period, with its worst calibration occurring for patients whose infections are detected one day after their admission to hospital, the time period that would be expected to be the most challenging to assess nosocomiality during.

## Discussion

We have presented NOSTRA, a new model that integrates both epidemiological and genetic data to give a posterior distribution over the potential infection origins of an individual. This new model represents the conceptual unification of the HOCI tool [6], for nosocomiality assessment, and the A2B model [11], for infection source identification. As all the terms in our model are mathematically tractable, we get the posterior distribution over sources in analytic form, allowing us to avoid any numerical integration and keep runtime low. There are few published models designed to estimate whether an infection is nosocomial [6–8], and

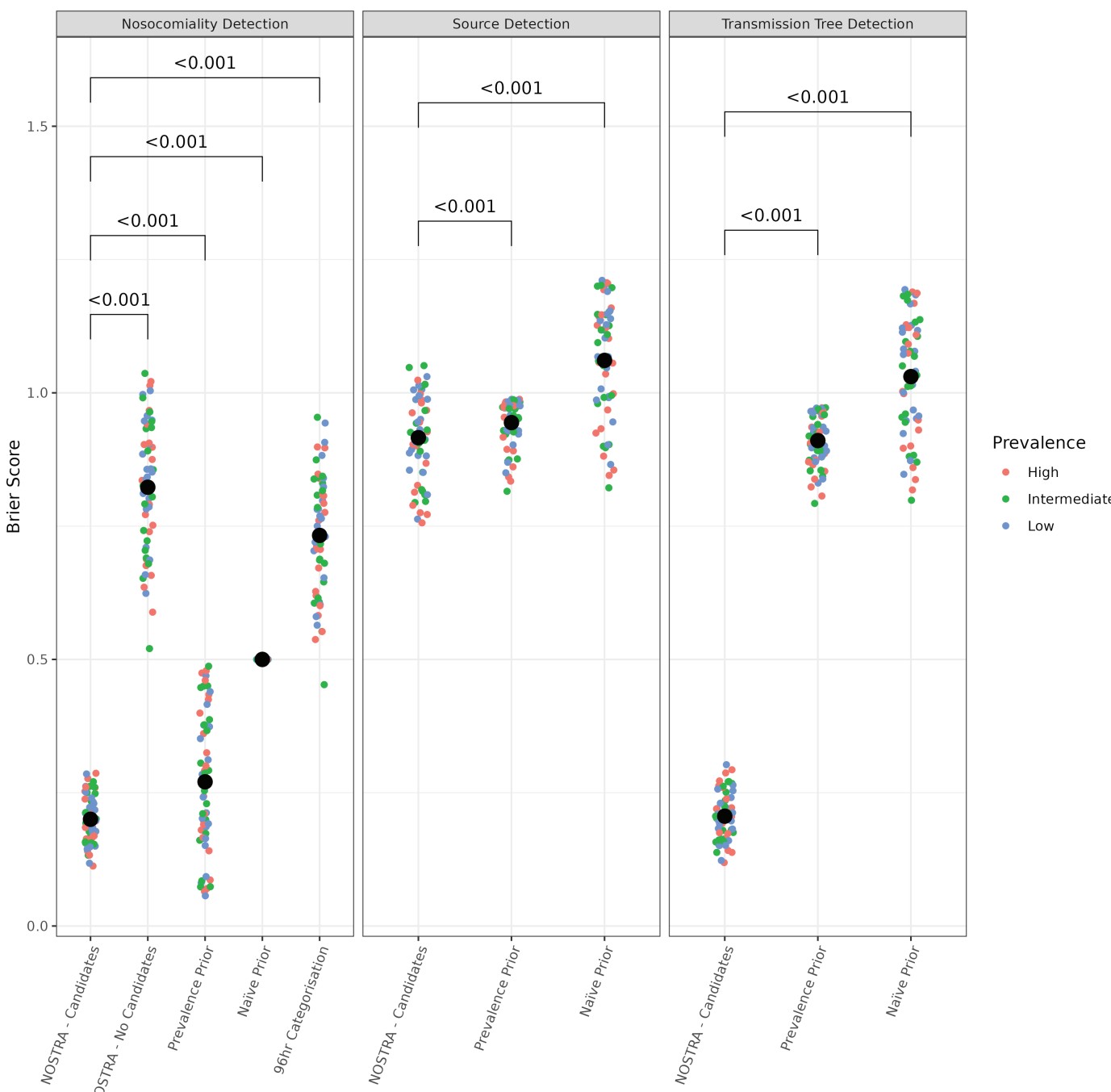

**Fig 3. The calibration of NOSTRA versus references as measured by Brier score.** The calibration of NOSTRA versus references as measured by Brier score for nosocomiality assessment (left), source identification (middle), and transmission chain identification (right). Low scores indicate better calibration. Points are coloured by the prevalence used in that simulation (see methods). The large black points correspond to the mean across simulations. NOSTRA, Candidates is the NOSTRA model run with a full set of candidate individuals and all data. NOSTRA, No Candidates is the NOSTRA model run with no candidate individuals using Eq 17. Prevalence Prior sets the prior probability of nosocomiality to the true probability of nosocomiality in the dataset. Naïve Prior sets the prior probability of community infection to 0.5 and the probability of every source within the hospital to $\frac{1}{2n+2}$, where $n$ is the number of candidate individuals in the hospital. 96hr Categorisation assigns a nosocomiality probability of 0 to anything detected in the first 96 hours post-admission and a nosocomiality probability of 1 to everything else. The backets show the Bonferroni-corrected $p$-values of the one tailed paired Wilcoxon signed rank test that the Brier score of the NOSTRA model run with candidates is lower than each reference.

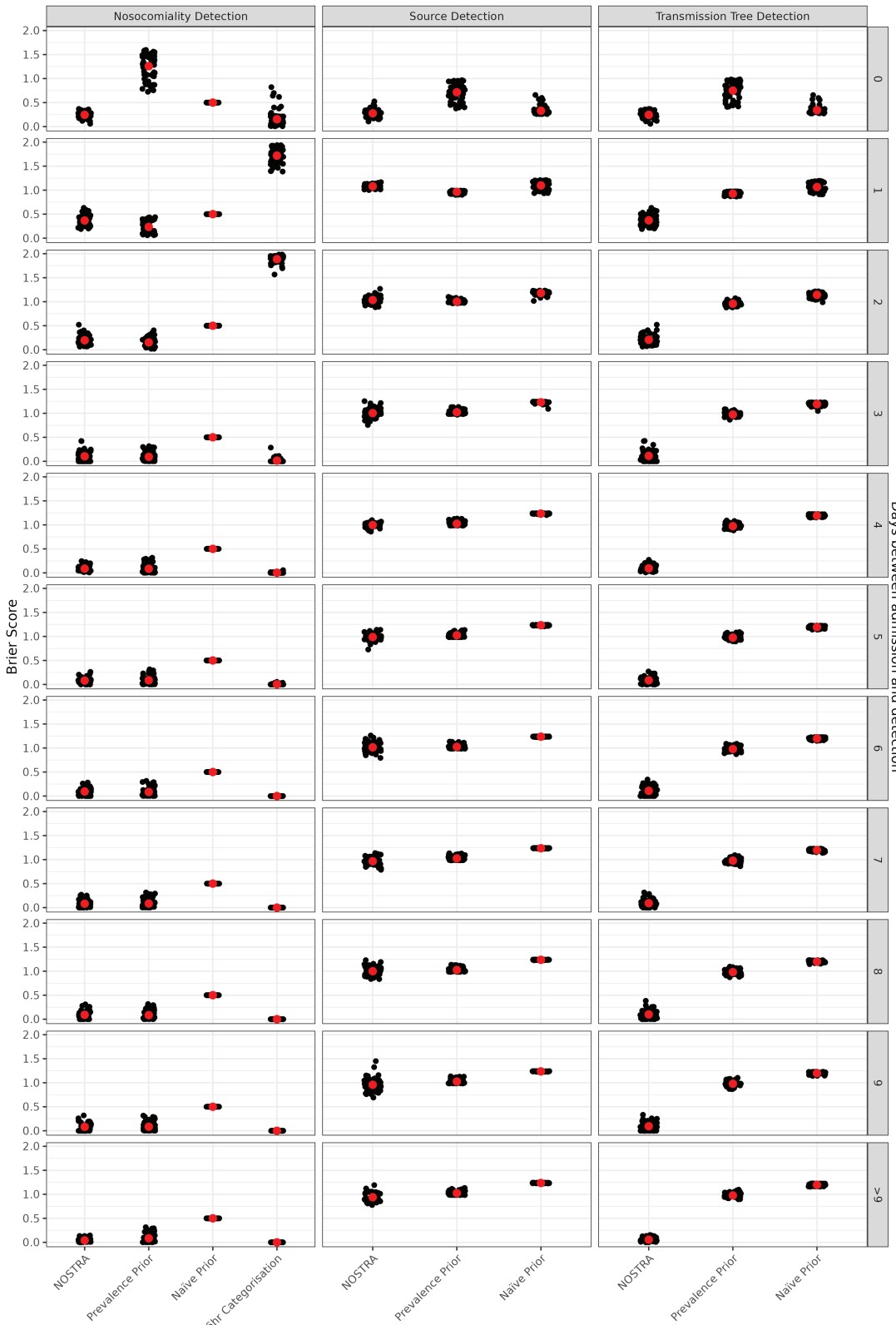

**Fig 4. The calibration of NOSTRA versus references as measured by Brier score by the between admission and detection of infection.** The calibration of NOSTRA versus references as measured by Brier score for nosocomiality assessment (left), source identification (middle), and transmission chain identification (right) by the time between admission and detection of infection. Low scores indicate better calibration. The large red points correspond to the mean across simulations. NOSTRA -

Candidates is the NOSTRA model run with a full set of candidate individuals and all data. NOSTRA - No Candidates is the NOSTRA model run with without any candidate individuals, using Eq 17. Prevalence Prior sets the prior probability of nosocomiality to the true probability of nosocomiality in that simulation run. Naïve prior sets the prior probability of nosocomiality to 0.5. 96hr Categorisation assigns a nosocomiality probability of 0 to anything detected in the first 96 hours post-admission and a nosocomiality probability of 1 to everything else.

to our knowledge no models previously were designed to jointly estimate the probability of nosocomiality and infection sources within the hospital.

NOSTRA provides probabilities, not dichotomised answers to the question of whether an infection is nosocomial. An important question that we have not attempted to answer in this work is how healthcare workers should interpret these probabilities for action. We believe that labelling a particular infection as "nosocomial" (or otherwise) on the basis of the model probability is, in effect, a policy decision. As such, this should be taken with reference to the potential costs and benefits of any action that would be taken in response to that decision, and how these would change if the classification were wrong. Therefore, it is important that thresholds for labelling be determined by individuals embedded within healthcare systems who are aware of the economic consequences of the decision for that particular system, and it is implausible that any one-size-fits-all approach will be appropriate.

The results from the CUH dataset give us some evidence on the kinds of data that may be useful to collect for the assessment of nosocomiality, irrespective of the method that is to be applied. In this case, the genetic data were very informative, as it allowed high probability candidates within the hospital to be identified. This suggests that routine sequencing of hospital pathogens may allow better assessments of nosocomiality, as well as being useful for tracking transmission networks. The waiting time between admission and onset provides a great deal of information about the location of the infection. If this waiting time is almost always less than five days, and the patient was admitted six days ago, then assessed probability of nosocomiality by NOSTRA is always going to be high, even if there are no genetically consistent identified candidates. Therefore, the genetic and location information are going to have the most impact on the assessed probability of nosocomiality in "difficult" cases. That is, when the time of onset is consistent with both hospital and community transmission, either because the observed time is right in the middle of the waiting time distribution, or because the waiting time is highly variable. Thus, we theorise that hospital sequencing of isolates may be especially valuable for pathogens with highly variable incubation times.

The simulation results we present help make clear NOSTRA's strengths and weaknesses. Namely, the Brier scores indicate, for the agent-based SARS-CoV-2 hospital infection model we used, in absolute terms NOSTRA performed well at estimating whether an infection was nosocomial and identifying the transmission chain the true source was in, but comparatively poorly at identifying the precise identity of that source. If the model has been well parameterised for the specific pathogen under study, in real usage we expect a decrease in NOSTRA's performance relative to what was seen here, because the way that the genetic data was simulated closely matches NOSTRA's assumptions. While the simulation model generates transmission trees, and hence does not conform to the assumption of independence between different sources we made in the derivation of the likelihood, the mutation process on those transmission trees follows the model that NOSTRA uses to calculate the genetic likelihood, with the only difference being the lack of additional noise from sequencing. The complexity of the substitution process in reality will lead to a degradation in performance of this part of the model. However, as the rest of the data were not simulated under NOSTRA's assumptions,

> **Box 1: Caveats for the usage of NOSTRA**
>
> 1. NOSTRA requires well defined generation and incubation times of the pathogen under study
> 2. NOSTRA requires knowledge about the mutation rate and effective population size of the pathogen under study
> 3. NOSTRA requires that the pathogen is directly transmitted between individuals
> 4. NOSTRA should not be applied to pathogens with high levels of within host variation
> 5. NOSTRA should not be applied to focal individuals who are asymptomatic
> 6. NOSTRA should not be applied to pathogens where indirect transmission is believed to be important
> 7. Infection candidates with high posterior probability of sourcehood should be considered linked infections rather than definite sources

and thus NOSTRA has already shown some robustness to violations here, we expect less loss of performance in the other components of the model.

NOSTRA is currently the only real option for joint estimation of nosocomiality and infection sources, it has some important caveats for use. We summarise the caveats potential users should take into account in Box 1.

A first caveat regards our handling of the genetics of the pathogen. We implicitly assume that there is a single genetic type at any time in each host. Explicitly we assume that given data for $A_z$ and $B$, when $A_z$ was the infection source, the time of the most recent common ancestor was the point of infection, $t_i^B$. This is only true if $A_z$ had no within host variation in its pathogen. If there is within host variation, this actually represents a lower bound on the time to the common ancestor, due to the potential for incomplete lineage sorting [27]. It is unlikely that this is likely to cause a large issue in most cases in which we envision that NOSTRA would be applied, given that the short generation times of most respiratory viruses means that the upper bound on the time to the most recent common ancestor is likely to be close to the lower bound. However, in infections with long generation times, where large amounts of within host diversity may be generated and maintained, this assumption may have a large impact, causing the number of expected SNPs between the infector and infectee to be underestimated. To account for this, a more complicated model allowing for pathogen diversification within hosts after infection would be required. Thus, we advise that users should not use NOSTRA for pathogens with long generation times.

A second caveat is about our handling of missing data. While NOSTRA can run with all data other than the focal individual's onset time being missing, we are making strong assumptions about the nature of that missingness in order to do this. We assume that the data is missing completely at random. That is, that the missing data is a random subset of the full data and it being missing is independent of both the values of observed and unobserved data. The degree to which this assumption holds is likely to depend on the specifics of the pathogen and hospital that this is being applied to. For example, samples with low pathogen load are known to fail sequencing more frequently than those with high load, so in a case where failed sequences are not reattempted, a missing genetic sequence may be indicative of someone early or late in their infection course, or it may simply be that their sample was not sent for sequencing for an unrelated reason. A full understanding of the data providence is necessary

to assess whether this assumption is reasonable, and as such whether the model is appropriate to be applied in the case of the user's specific missing data.

A type of missing data that is worth commenting on specifically is asymptomatic carriage. In the case of asymptomatic carriage symptom onset times will be unobserved (and, indeed, undefined). While we did this for illustrative purposes with the CUH data, we strongly advise that, in practice, NOSTRA should not be applied to focal individuals who are asymptomatic. This is because there is no well-defined distribution that describes the waiting time between admission and detection for asymptomatic individuals. It is entirely a function of the testing strategy applied by the healthcare facility. This does not mean that NOSTRA is inapplicable to pathogens where asymptomatic carriage is an important factor in the epidemiological dynamics, as long as the focal individual is symptomatic. If one is willing to accept the missing completely at random assumption, candidate infectors who are asymptomatic can be included in the model with their onset times as missing data, this allows the genetics of their pathogen to still inform the likelihood. If one is not willing to make that missingness assumption, these individuals can simply be ignored, which is equivalent to including them in the "Hospital" source.

Another important kind of missing data is that of unrecognised infector candidates in the hospital. Our "Hospital" infection source allows for the true infector in the hospital to be unidentified. However, pathological behaviour is likely to occur if the true infector is unidentified, but there is a consistent infector in the set of identified candidates, likely from the same transmission chain. Under this circumstance, the consistent infector will be likely to be assessed as having high posterior probability at the expense of the "Hospital" source. Given this, the "Hospital" source should be considered a guard against the possibility that there are no likely identified candidates, but the timings of infection strongly suggest nosocomiality. This means that, in cases where the "Hospital" source is identified as the most probable, it does indicate that the detected infection genuinely is unrelated to anything in the candidate set. Therefore, while NOSTRA is not designed for this, with appropriately designed candidate sets (i.e. excluding a class of interest), NOSTRA could potentially be used to explore whether different classes of individuals are involved in within hospital outbreaks. However, we would advise further validation studies be performed before this were attempted.

Related to missing the true infector and inappropriately assigning an individual from the same transmission chain as the source, is the case where the data are too weak to correctly identify which individual within a transmission chain is the source, even if they are in the set of candidate individuals. We believe that this explains NOSTRA's relatively poor performance at identifying the true source, but very good performance at identification of sources within the same transmission chain in the simulations. Given a transmission chain that takes place over a week or two in a single ward, the number of SNPs separating the cases will be small and all detected cases are likely to have been co-located. In this case, NOSTRA commonly ends up placing similar probability over all cases. Unlike the case of the true infector being missing and another individual in the transmission chain being assigned high probability, this is not unintended behaviour, as there genuinely is uncertainty about the true source which cannot be resolved by the data. However, this does mean that in many cases a single source will not be identifiable. Given these issues, and the high accuracy achieved for the detection of individuals within the transmission chain seen in the simulations, we advise users to consider the high probability infection sources within the hospital as likely linked cases, rather than the definite source of the infection.

A third thing that users should take into account when using NOSTRA is its prior. A prior that is uniform across infection sources should be avoided, as this has the unfortunate

consequence that the induced prior on whether the infection is nosocomial becomes a function of the number of candidate individuals within the hospital. For this reason, in actual usage, we advise firstly defining a prior probability that the infection occurred in the community, and then distributing the rest of the probability equally over the hospital-associated infection components. Our default throughout this work has been to set this prior probability of community infection to 0.5, and this performed well in our simulations, but an informed prior based on knowledge of the approximate frequency of nosocomial infections in the facility where NOSTRA is being used will lead to better performance in general.

The required complexity of NOSTRA to model the multiple data sources that can be inputted to it means that it has higher requirements for prior knowledge about the biology of the pathogen of interest than many other similar epidemiological models. Specifically, studies must have been performed estimating an effective population size for the pathogen in the recent past, in order for the genetic likelihoods to be calculable. This is not a large problem for well-studied infections of the kind that we believe NOSTRA is likely be applied to (e.g. SARS-CoV-2, RSV and influenza) where phylogenetic studies are regularly being performed and values will be accessible in the literature. However, it does represent a limitation with respect to understudied or novel pathogens, where effective population size estimates may not be available.

Another potential limitation is that to ensure analyticity of the posterior distribution, allowing its direct calculation, we had to make strong independence assumptions between the different data types. The assumption of independence between the genetic data and the epidemiological data for $A_z$ and $B$, when $A_z$ was the infection source, being one notable for example. In reality, there will be complex interrelations between these two data types, given that $B$'s epidemiological data and the genetic distance from the isolate from $A_z$ should depend on $A_z$'s epidemiological data through its influence on the infection time of $B$. This could lead to over- or under-estimation of the probability of $A_z$ being the infection source of $B$ depending on the precise combination of the data.

One final limitation relates to the use of the coalescent to model "unrelated" genetic sequences. The form of the coalescent we use makes two assumptions that might be problematic. Firstly, that there is no selection. Over short periods of time, where there is one dominant genetic type, this might be approximately true, but over longer time periods, it will definitely not be. Secondly, that there is no population structure. It is likely that there will be some degree of spatial structuring, and that the sequences in the hospital will be more closely related than would be the case if they were drawn at random from the entire population. This means that $T_{MRCA}^{A_z,B}$ is likely to have a mean that is too high, i.e. that the time to coalescence would be shorter than would be expected for two sequences drawn at random, and consequently, the expected number of SNPs between the isolates would be overestimated. Both of these issues could potentially be resolved by modifying the form of the coalescent used, likely at the cost of more prior knowledge being required, but that goes beyond the scope of this work.

## Conclusion

We believe that NOSTRA represents a step forward in data integration for nosocomial infection detection. Our tool provides the probability that an infection is nosocomial as well as the probability that certain given candidate individuals were linked to the infectee, something that was not previously available. We have reached the point that there are now multiple models that purport to assess nosocomiality in the literature, but we are limited by the absence

of datasets where both the truth is known and the answer is non-trivial, so the accuracy of the assessments could not be quantified or compared. The simulation tools used in the evaluation here would likely be useful for more general comparisons between these methods, so that clinicians and medical statisticians can choose to implement the model that they would expect to perform best for their specific scenarios.

## Supporting information

**S1 Table. The parameter sets used for the hospital simulations.** Parameters starting with b are transmission rates used in the model. bP2P is the within bay transmission rate between patients. bP2P_hosp is the indirect between patient hospital transmission rate. bH2P is the healthcare worker to patient transmission rate. bP2H is the transmission probability from patients to healthcare workers per timestep. bH2H is transmission probability from healthcare workers to other healthcare workers per timestep. bH2H_hosp is the indirect between healthcare worker hospital transmission rate. commScale is the scale of community acquisition rate for HCWs. See the supplementary materials of Evans et al. [24] for a full description of the model.
(XLSX)

**S1 Fig. The concentration of posterior mass on specific sources as data is added.** This figure shows an example NOSTRA run from one simulated individual from the simulation analyses. Potential candidate infectors for individual are labelled 1 to 16. The high of each bar corresponds to the posterior mass placed on that infection source. The top left panel shows the prior probabilities of each infection source. The top right panel shows the posterior probabilities of each infection source after admission and onset times are added (dark) and the prior probabilities of each infection source (light). The bottom left panel shows the posterior probabilities of each infection source after admission times, onset times, and location information are added (dark) and the posterior probabilities of each infection source after admission and onset times are added (light). The bottom right panel shows the posterior probabilities of each infection source after all data are added (dark) and the posterior probabilities of each infection source after admission times, onset times, and location information are added (light).
(TIFF)

## Author contributions

**Conceptualisation:** David J. Pascall, Theodore Gouliouris, Christopher J. R. Illingworth, Stefan G. Piatek, Oliver Stirrup, Ben Warne, Judith Breuer, Daniela De Angelis.

**Methodology:** David J. Pascall, Christopher Jackson, Stephanie Evans, Daniela De Angelis.

**Software:** David J. Pascall, Stephanie Evans, Stefan G. Piatek.

**Validation:** David J. Pascall.

**Formal analysis:** David J. Pascall, Stephanie Evans.

**Investigation:** David J. Pascall, Christopher Jackson, Stephanie Evans.

**Data curation:** David J. Pascall, Stephanie Evans, Ben Warne.

**Writing - original and draft:** David J. Pascall, Christopher Jackson, Theodore Gouliouris.

**Writing - review and editing:** David J. Pascall, Christopher Jackson, Stephanie Evans, Theodore Gouliouris, Christopher J. R. Illingworth, Stefan G. Piatek, Julie V. Robotham, Oliver Stirrup, Ben Warne, Judith Breuer, Daniela De Angelis.

**Visualisation:** David J. Pascall.

**Supervision:** Julie V. Robotham, Daniela De Angelis.

**Funding acquisition:** Judith Breuer, Daniela De Angelis.

**Project administration:** Judith Breuer, Daniela De Angelis.

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
