## [Decision Letter · Decision Letter 0]

1 Jul 2024

Dear Dr. Pascall,

Thank you very much for submitting your manuscript "The NOSTRA model: coherent estimation of infection sources in the case of possible nosocomial transmission" for consideration at PLOS Computational Biology.

As with all papers reviewed by the journal, your manuscript was reviewed by members of the editorial board and by several independent reviewers. In light of the reviews (below this email), we would like to invite the resubmission of a significantly-revised version that takes into account the reviewers' comments.

Apologies for the multiple delays in providing a decision on this manuscript. As you will see from the reviewer comments, there are some common themes that will be important to address in your revised manuscript. In particular, both Reviewers 1 and 3 note the need to validate the model against simulated data. This is also important from a reproducibility standpoint, as it is noted that the original data cannot be shared due to privacy concerns, but a simulated dataset can and should be provided to allow for model testing. The reviewers also note the importance of being clearer about the limitations of the approach, e.g. that it only applies to acute infections.

We cannot make any decision about publication until we have seen the revised manuscript and your response to the reviewers' comments. Your revised manuscript is also likely to be sent to reviewers for further evaluation.

Sincerely,

Virginia E. Pitzer, Sc.D.

Section Editor

PLOS Computational Biology

Virginia Pitzer

Section Editor

PLOS Computational Biology

Apologies for the multiple delays in providing a decision on this manuscript. As you will see from the reviewer comments, there are some common themes that will be important to address in your revised manuscript. In particular, both Reviewers 1 and 3 note the need to validate the model against simulated data. This is also important from a reproducibility standpoint, as it is noted that the original data cannot be shared due to privacy concerns, but a simulated dataset can and should be provided to allow for model testing. The reviewers also note the importance of being clearer about the limitations of the approach, e.g. that it only applies to acute infections.

Reviewer's Responses to Questions

**Comments to the Authors:**

Reviewer #1: This manuscript presents NOSTRA, a new model designed to estimate the probability of detecting a hospital acquired infection and the probabilities of the source being within a set of candidate individuals. Calculating these probabilities is set up as a Bayesian inference problem and thus requires specifying a prior and generates a posterior probability distribution. The authors demonstrate the utility of their method using previously published data collected during the early stages of the COVID-19 pandemic. They use data from one out of five hospital wards in the published dataset where patients were tested for COVID through targeted patient screening in wards with detected hospital-onset outbreaks. Data on patients includes movement, the genetic sequence of their infecting pathogen, their times of admission to hospital and symptom onset, and the movements and onset times of a set of n candidate infector individuals in the hospital. Some patients have associated genetic data while other patients are missing this information. The results demonstrate that probabilities of the infection source can be estimated, and that adding data sources improves their estimates. The authors are clear about assumptions and when NOSTRA should be and should not be used, and clearly define the purpose of their tool.

Critical Comment:

Overall, I believe the authors have developed a useful tool on solid mathematical and statistical foundations. My main concern/critique, which needs to be addressed in the revision, is that this paper completely lacks any analysis or demonstration that the uncertainty or biases around the estimates produced by the model are reasonable. I do not think the manuscript should be published until metrics of uncertainty are included. I think the most robust way to go about testing this new method would be to create simulated datasets with known infection origins and infectors and determine how often your model can predict the correct designation with a high probability. It would also be helpful if the authors could present credible intervals around each of their probability estimates and show how these intervals change if the patient has genetic data associated with their test. If there are too many estimates to put in the main text, the authors could add a supplement to their paper to house these results. Figures depicting uncertainty would greatly strengthen the paper.

Major Comments:

The Bayesian Analysis: Inference section does not offer enough detail. I think it would be helpful to any future user of your method to describe the algorithm you used to sample from your posterior (or suggest one) or state you are calculating this distribution directly. It is not clear what you are doing unless you look at your code.

The authors should offer guidance on how to interpret the probabilities. Is there a cutoff the authors recommend to designate, for example, if an infection is either nosocomial or not?

Overall, figures and figure captions need to be more informative:

First, I think it would be beneficial to have at least one figure that demonstrates the uncertainty around your estimates.

Figure 1 – Why is hospital and nosocomial included on the x-axis? What is the difference between hospital and nosocomial? Also, what is the “full model”? Please remind the reader in the caption what the figure represents.

Figure 2 – It would be helpful to have the panels labeled (e.g. A, B, etc.) and describe specifically what each panel represents. It seems like the very last panel in this figure is different (e.g. it is not a measure of change in probability). What results are shown in the last panel, and should it be a separate figure?

I do not think the authors adequately justify the importance of understanding the origins of infections in health care settings. I think this is a missed opportunity. For example, in line 6 the authors state: “These two scenarios have very different implications for decision making.” But then only follow this statement by explaining one of the scenarios (ward closure), but not both. I think additional impacts of identifying if an infection is nosocomial or not and the transmission chain are more than just ward closure. This is explained in line 46 with regard to COVID, but this seems like it could be generally applicable to many different types of infections (e.g. bacterial infections). I would like to see the authors add more detail in the first paragraph of the introduction explaining why this is generally important. (This could also be mentioned in the conclusion).

I am unable to run any of the code in the NOSTRA-model repository. If the data cannot be made available, the authors should produce a simulated dataset that a user can use to test the code. Additionally, there is no information about the repository in the Readme file. The code needs to be described in the Readme file in the repository.

Minor Comments:

Abstract - This is confusing: “example dataset from a real dataset”. It is not clear until reading the methods what the authors mean by “example dataset.”

Line 93 - The use of viral sequences in line 93 seems abrupt. For readers who are not familiar with genomic sequences, it would be helpful to expand and introduce this method with a bit more detail.

Consider moving some of the mathematical details about the likelihood section to an online supplement.

Reviewer #2: Thank you for the opportunity to review the manuscript by Pascall and colleagues that describes a Bayesian model that estimates whether an infection detected in hospital is of nosocomial origin, as well as the likelihood of being derived from a proposed group of source patients. The manuscript is well written and addresses a frequently encountered, but challenging problem of determining source of acquisition for infections detected in hospitalised patients. In practice, Infection Prevention & Control interventions often take a conservative and sometimes excessive approach when applied to outbreak control, though this is due to the difficulty in identifying the source in each case. In this manuscript, the authors provide a glimpse of what may be possible to glean from detailed genomic and epidemiological data and I believe would be of interest to healthcare administrators and those working in Infection Prevention & Control.

It should be noted that I have reviewed the manuscript from a clinician perspective as although I have some rudimentary knowledge of modelling, I do not have the detailed mathematical background to review the model in detail. I have included some comments below intended to help improve the readability and interpretation of the manuscript from a clinician perspective.

1. It would be helpful if Figures 1 and 2 could be presented in the same orientation to aid interpretation.

2. Could you please define the terms "nosocomial", "community" and "hospital", as used in the output of your model? The definition of nosocomial usually includes transmission from any individual in the healthcare environment, not just from the set of candidate individuals i.e. {A,H}. However, I understand your rationale to separate these as infections from "known positives" (I think what you have labelled "nosocomial") and from unrecognised sources in the hospital environment (I presume what you have labelled "hospital").

3. Transmission from individuals in the hospital that are outside of the set of candidate individuals are also relevant to healthcare-associated outbreaks, and could be from staff, visitors, or other unrecognised patient cases, each with a different outbreak control intervention. This is alluded to in the discussion, but consider expanding the discussion on the potential for monitoring number of healthcare-associated cases that are related to "known positives" vs those from "unrecognised sources" in the hospital.

4. Consider a box listing key caveats of implementing NOSTRA e.g.

- requires infection with short, well-defined incubation periods and generation times (given the dependence on this to infer nosocomiality/hospital-acquisition)

- requires a reasonable proportion of the samples in the hospital to be sequenced

- better suited to pathogens without significant within-host variation

Reviewer #3: The manuscript by Pascall et al. describes a new model (NOSTRA) for reconstruction of transmission of infections in the hospital environment.

The NOSTRA model builds upon two existing models developed by the group: the A2B model that determines who infected whom in hospital outbreaks, and the HOCI model that assesses the probability that an infection was nosocomial. The authors report that the originality of this model is that it would be the first to simultaneously assesses whether an infection is nosocomial in origin (versus community-acquired) and who the likely infector is.

This research question addressed by the authors is both important and relevant, for the modelling community but also for infection prevention and control professionals working in hospitals. The authors should be lauded for the thoroughness of their approach and the significant amount of work that was put into this. Additional strengths of the manuscript include the use of ward location data for patients, which are, to the best of our knowledge, not used in publicly available approaches. Finally, the manuscript is well-written and easy to follow conceptually.

Despite these strengths, there are major limitations that we would like to highlight.

The scope of this approach is not useful for all nosocomial infections – it would only be useful for acute viral respiratory infections. In the case of bacterial infections, even in outbreak situations, there exists the problem of asymptomatic carriage which is not accounted for in the model. Please ensure that this is made clear in the title and main text, or explain how asymptomatic carriage (with or without subsequent infection) are accounted for.

The premise that simultaneous determination of “nosocomiality” and source attribution would be better than sequential assessment is not in itself intuitive. Can the authors please expand on what the added value of this approach would be? Also, it would be interesting to perform a comparison of the characteristics of the NOSTRA model and other existing approaches that could be used to address the same problem.

While we agree with the authors’ premise that the definitions of nosocomial infection are somewhat arbitrary (and perhaps most useful for surveillance purposes), we are left wanting more regarding how useful this model would be in practice. Presumably, there are some cases where the attribution of “nosocomiality” are straightforward. For what proportion of cases do the authors expect there to be a benefit?

A major limitation is the lack of assessment of the model’s performance in a simulated outbreak, where the “ground truth” is known. A comprehensive simulation study is essential to understand the model's behaviour under different conditions, as commonly done with novel outbreak reconstruction tools. Evaluation typically involves diverse conditions like population sizes, R0 values, sampling coverage, mutation rates, and generation time/serial interval distributions.

Another limitation lies in the fact that healthcare workers do not seem to be included in the model. It is well-established that healthcare workers are key drivers of nosocomial outbreaks of acute respiratory viral infections.

We disagree with the authors’ premise that “the appropriate infection control response depends on whether an infection truly is nosocomial”. The unexpected occurrence of a respiratory viral infection in a patient who was not symptomatic of that infection on admission would trigger the same investigations and response whether the infection is acquired in the hospital or the community.

Additional comments

• It may be worth assessing the model's robustness to changes in population size, as suggested in a related study focusing on detecting imports from a genetic coalescent model (https://www.ncbi.nlm.nih.gov/pmc/articles/PMC9805578/).

• How are asymptomatic cases handled?

• How does not accounting for missing links/generations affect the inference about “nosocomiality” or the source?

• “Waiting time” (line 147) should be replaced by “incubation period”.

**Have the authors made all data and (if applicable) computational code underlying the findings in their manuscript fully available?**

Reviewer #1: **No: **The authors state they cannot share the data because it is not de-identified. I think the authors should de-identify the data and make the data publically available.

Reviewer #2: Yes

Reviewer #3: **No: **

PLOS authors have the option to publish the peer review history of their article (what does this mean?). If published, this will include your full peer review and any attached files.

Reviewer #1: No

Reviewer #2: **Yes: **Jason Kwong

Reviewer #3: **Yes: **Mohamed Abbas
---

## [Decision Letter · Decision Letter 1]

16 Dec 2024

PCOMPBIOL-D-23-01784R1

The NOSTRA model: coherent estimation of infection sources in the case of possible nosocomial transmission

PLOS Computational Biology

Dear Dr. Pascall,

Thank you for submitting your manuscript to PLOS Computational Biology. After careful consideration, we feel that it has merit but does not fully meet PLOS Computational Biology's publication criteria as it currently stands. Therefore, we invite you to submit a revised version of the manuscript that addresses the points raised during the review process.

Please submit your revised manuscript within 30 days Feb 15 2025 11:59PM. If you will need more time than this to complete your revisions, please reply to this message or contact the journal office at ploscompbiol@plos.org. Please include the following items when submitting your revised manuscript:

We look forward to receiving your revised manuscript.

Kind regards,

Virginia E. Pitzer, Sc.D.

Section Editor

PLOS Computational Biology

Virginia Pitzer

Section Editor

PLOS Computational Biology

**Additional Editor Comments:**

Apologies for the delay in obtaining the reviews for the revised manuscript. As you will see, Reviewer 3 is satisfied with the changes that have been made and only couple minor comments. However, Reviewer 1 raises a number of issues that require further clarification, although I think the changes needed are fairly minor. In particular, I think it would be helpful to explicitly state that you are calculating the posterior probability of the different categorical sources of infection analytically rather than sampling from the posterior using MCMC, for example, at the beginning of the "Bayesian Analysis: Inference" section (as you do in lines 295-296, but I think this could be moved up before the equation for the full posterior likelihood and written a little more clearly).

**Journal Requirements:**

1) We notice that your supplementary Tables are included in the manuscript file. Please remove them and upload them with the file type 'Supporting Information'. Please ensure that each Supporting Information file has a legend listed in the manuscript after the references list.

2) Please ensure that the funders and grant numbers match between the Financial Disclosure field and the Funding Information tab in your submission form. Note that the funders must be provided in the same order in both places as well.

**Reviewers' comments:**

Reviewer's Responses to Questions

**Comments to the Authors:**

Reviewer #1: This manuscript presents a revision of the NOSTRA, a new model designed to estimate the probability of detecting a hospital acquired infection and the probabilities of the source being within a set of candidate individuals. The authors have added a model validation analysis to help address the predictive performance of their model that calculates the probability of nosocomilality, but these sections still need work. The results from this new analysis are not clearly stated and these sections need more revision. In addition, I do not think the authors have adequately addressed my concern about the Bayesian Analysis: Inference section having enough detail. I still do not understand why the authors cannot quantify uncertainty of the estimated probabilities.

Major comments:

The Model section still needs more revision. It is hard to follow. The use of semicolons followed by lists in the paragraph beginning on line 113 I think are particularly for confusing the reader. I think it would be helpful to explain very clearly in the first few sentences the possible sources of infection in a more logical order. 1) Outside the hospital, C. 2) Non-patient source but in the hospital, H. 3) The A’s. Why is H the n+1 source and C the n+2? Can C be the first source, H the second source, and the A’s be 3 to n?

The probability statement in line 125 is P(S|X)=P(A1,…,An, H, C|X). Why is this the probability that B’s infection can from “each of the n+2 sources”? I do not follow this logic and I think this should be further explained. Why is your prior, P(S), not included in your probability statement?

The Bayesian analysis: Inference section beginning on line 293 describes how to calculate the posterior probability from the analytical solution. This is related to my initial comment about the lack of uncertainty quantification. What is being “inferred” from equation 16 and I still do not understand why the authors cannot quantify the uncertainty in their calculated probabilities.

In many places the manuscript is confusing, and the model is hard to follow. For example, in lines 358-366 it would be much clearer if the authors simply stated something like, “we used Brier scores with the original definition of the loss function, where these scores measure the accuracy of probabilistic predictions.”

The model validation results beginning on line 416 would benefit from adding a summary of the resulting Brier scores and how the different models compare based on these scores.

Line 488 in the discussion states, “The simulation results we present help make clear NOSTRA’s strengths and weaknesses.” The strengths and weaknesses are not clear. These should be quantified in the results (this is my suggestion of adding a summary of Brier scores and how the models differ in these scores) and this needs to be explained in the discussion.

In line 502 in the discussion the authors state, “However, as the rest of the data were not simulated under NOSTRA’s assumptions, we expect little loss of performance in the other components of the model.” Why do you expect little loss of performance? How can you make this claim without testing this with simulated data?

In line 505 the authors use the term “joint estimation.” Joint estimation of what? In my comment about the Bayesian inference section, I ask what is begin inferred. The authors need to make this clarification throughout the manuscript.

Table 2 should be moved to the model section of methods and not be presented in the discussion.

Reviewer #3: The authors should be congratulated for greatly improving the manuscript, especially with regards to the addition of the simulation study. Our concerns were appropriately addressed.

Minor comment:

- on page 12 of the manuscript, in the sentence "those being detected on the day they are admitted being easily identifiable as nosocomial" - shouldn't it be community-acquired?

- should the relatively long run time of the model (4 days) be included as a potential limitation to routine implementation in a clinical setting?

**Have the authors made all data and (if applicable) computational code underlying the findings in their manuscript fully available?**

Reviewer #1: Yes

Reviewer #3: Yes

PLOS authors have the option to publish the peer review history of their article (what does this mean?). If published, this will include your full peer review and any attached files.

Reviewer #1: No

Reviewer #3: No

**Figure resubmission:**
---

## [Editor Report · Decision Letter 2]

10 Mar 2025

Dear Dr. Pascall,

We are pleased to inform you that your manuscript 'The NOSTRA model: coherent estimation of infection sources in the case of possible nosocomial transmission' has been provisionally accepted for publication in PLOS Computational Biology.

Best regards,

Virginia E. Pitzer, Sc.D.

Section Editor

PLOS Computational Biology

---

## [Editor Report · Acceptance letter]

PCOMPBIOL-D-23-01784R2

The NOSTRA model: coherent estimation of infection sources in the case of possible nosocomial transmission

Dear Dr Pascall,

I am pleased to inform you that your manuscript has been formally accepted for publication in PLOS Computational Biology. Your manuscript is now with our production department and you will be notified of the publication date in due course.

With kind regards,

Zsofia Freund
